# Evaluating feature extraction in ovarian cancer cell line co-cultures using deep neural networks
Osheen Sharma [1] ✉, Greta Gudoityte[1], Rezan Minozada[1], Olli P. Kallioniemi[1,2], Riku Turkki[2], Lassi Paavolainen [2,3] & Brinton Seashore-Ludlow [1] ✉

Single-cell image analysis is crucial for studying drug effects on cellular morphology and phenotypic changes. Most studies focus on single cell types, overlooking the complexity of cellular interactions. Here, we establish an analysis pipeline to extract phenotypic features of cancer cells cultured with fibroblasts. Using high-content imaging, we analyze an oncology drug library across five cancer and fibroblast cell line co-culture combinations, generating 61,440 images and ~170 million single-cell objects. Traditional phenotyping with CellProfiler achieves an average enrichment score of 62.6% for mechanisms of action, while pre-trained neural networks (EfficientNetB0 and MobileNetV2) reach 61.0% and 62.0%, respectively. Variability in enrichment scores may reflect the use of multiple drug concentrations since not all induce significant morphological changes, as well as the cellular and genetic context of the treatment. Our study highlights nuanced drug-induced phenotypic variations and underscores the morphological heterogeneity of ovarian cancer cell lines and their response to complex co-culture environments.

High-content imaging (HCI) coupled with automated image analysis has emerged as a powerful, systematic method to associate cellular phenotype with a particular treatment or experimental condition[1–3]. These tools shed insight into fundamental biological processes and the small molecules or genes that modulate them, which are central aspects of both cell biology and drug discovery. Indeed, recent studies have demonstrated that morphological profiling provides a more nuanced description of treatment response[4,5], improving our understanding of the activity of small molecules and the effects of genetic variants or perturbations[6–9]. One major advantage of HCI is the ability to study the cellular response to treatment in an unbiased manner at scale. However, the volume and complexity of the data generated by HCI present key challenges in not only data storage but also efficient image analysis and downstream biological interpretation of results, which need to be addressed to truly leverage this capacity[10].

Feature extraction – transforming images into representations of cellular phenotype – is a key step in the image analysis process. Traditionally, this has been performed using CellProfiler (CP) or similar software[11,12], where feature extraction is handcrafted for each dataset[2,13,14]. Advances in deep learning, on the other hand, provide a unique opportunity to enable data-driven feature extraction with the potential to improve the efficiency and robustness of this process despite potentially confounding factors, such

as multiple cell lines, plates, or batches[15–17]. Recent studies have highlighted the advantages of deep learning-based feature extraction over traditional methods[17–20]. Within the area of neural networks (NN), features are derived using two different approaches. The first involves using pre-existing models that have been trained on extensive datasets, such as ImageNet[21–23], which is widely recognized in computer vision research[24]. These models have learned intricate details from a diverse range of natural images, making them a valuable resource for prediction and classification tasks. Alternatively, models can be trained specifically on fluorescence microscopy images using techniques like weakly supervised or self-supervised learning, allowing for the direct extraction of relevant features from specific research domains and offering deeper insights into cellular morphology and behavior[22,25,26]. While pre-trained models offer a robust foundation, training models on domain-specific data ensures a more tailored understanding of the complexities within the specific research field. Notably, these approaches can reduce human intervention, enhance performance in downstream analysis, and improve computational efficiency.

Since previous research[27–29] has focused on studying the phenotypic response of perturbations on individual cancer cell lines (CCLs), in this paper, we focus on extracting morphological features from fluorescent images of 2D co-culture assays containing both cancer cells and fibroblasts.

[1]Department of Oncology-Pathology, Karolinska Institutet, Science for Life Laboratory, Stockholm, Sweden. [2]Institute for Molecular Medicine Finland (FIMM), HiLIFE, University of Helsinki, Helsinki, Finland. [3]iCAN Digital Precision Cancer Medicine Flagship, University of Helsinki, Helsinki, Finland.
✉e-mail: osheen.sharma@ki.se; brinton.seashore-ludlow@ki.se

Our overall goal is to investigate how interactions between both cell types alter the response to chemical perturbations. To achieve this, we first need to identify a scalable method of feature extraction for individual cell types, which will enable us to interrogate more complex model systems. Moreover, we aim to assess the generalizability of these analysis methods across different CCLs and co-culture conditions. Here, we compare two feature extraction methods, CP and NN, across a dataset[30] including five different co-culture conditions tested against 528 compounds[31]. We make this unique dataset available to the community for future improvements. We focus specifically on morphological changes in the cancer cell component of the co-culture and adopt a weak auxiliary biological matching task similar to Moshkov et al.[18,32], to evaluate the ability of the two feature extraction methods to recover phenotypic outcomes due to compound treatment. Interestingly, we find that inherent cell line genetic and morphological profiles impact both feature extraction methods. Overall, these results provide insight into the future design and application of CP and NN-based feature extraction for the rapid biological interpretation of large-scale HCI datasets.

## Results

### Cancer cells co-cultured with fibroblasts have distinct morphological phenotypes

To study feature extraction methods in complex cellular settings, we first generated a 2D co-culture dataset using five different combinations of ovarian cancer and fibroblast cell lines (Table 1). To distinguish different cell types, antibodies were used to mark vimentin in fibroblasts and cytokeratin 8/18 (CK8/18) in epithelial ovarian CCLs (Fig. 1a, Supplementary Fig. 1). The co-cultures were treated with 528 drugs (Fig. 1b) at five concentrations followed by HCI (details in the "Methods" section). The compound library used has detailed annotations for known targets of each small molecule (Supplementary Table 1), providing comprehensive information about the intended mechanism of action (MOA). Before investigating the morphological profiles of the treated cells, we evaluated the technical quality across all assay plates by computing the Z' score using cancer cell count (retrieved from Hoechst staining) as the assay readout. The results consistently demonstrated scores greater than 0.5, indicating a clear separation between positive (benzethonium chloride (BzCl)) and negative control (dimethylsulfoxide (DMSO)) wells (Supplementary Table 2), and high technical quality.

Since our goal is to study drug-induced morphological changes in specific cell populations, it is essential to first accurately segment the individual cell objects belonging to the two populations. Here, we focus on methods to study the cancer cell population, using a deep learning-based segmentation technique Cellpose[18] (Supplementary Fig. 2) to identify individual cancer cells after image preprocessing. The resulting segmentation mask was utilized alongside the images from the three channels, CK8/18, vimentin, and Hoechst, for CP feature extraction. For the baseline NN approach, initially, a bounding box located at the centroid of each Cellpose object was used to extract features from three channels: CK8/18, vimentin, and Hoechst within the bounding box. Fig. 1c, d illustrates the feature extraction and evaluation process using both CP and NN. For NN feature extraction, we first evaluated three pre-trained baseline models: EfficientNetB0, MobileNetV2, and ResNet50. Features were extracted directly from the intermediate convolutional

layers of these networks without additional training, leveraging their pre-trained weights.

As an initial comparison of the morphological representations from the two feature extraction methods, we first examined their ability to distinguish the different CCLs. Based on results from previous studies, we expected each CCL to exhibit a distinct morphological profile[33,34]. In addition, we hypothesized that cancer-fibroblast interactions may impact cancer cell morphology, such that the same CCL cultured in the presence of different fibroblasts could have a distinct morphological fingerprint. For the CP feature extraction workflow, only features within the Cellpose mask of the cancer cells were evaluated. In contrast, for the initial pre-trained baseline NN workflow, the bounding box region is not masked and may include areas of neighboring cells as well. UMAP dimensionality reduction on the aggregated features from negative control (DMSO) wells of the cancer cell population shows five distinct clusters, each corresponding to specific co-culture conditions using both feature extraction methods (Fig. 2, PCA plots in Supplementary Fig. 3). Even when using the same CCLs in co-culture with different fibroblasts (e.g., KB and KW), distinct clusters were observed using both methods. This finding emphasizes the impact of fibroblasts on the morphology of the CCL in co-culture conditions, potentially providing a method to extract valuable insights into the cell-cell interactions from these models. Additionally, the close proximity of all the negative control wells from each cell line suggests experimental consistency across assay plates (Supplementary Figs. 4 and 5).

### Using masked single-cell objects improves representation learning

Next, we wanted to investigate the ability of CP and the pre-trained NN to recover phenotypic outcomes from compound treatment in each co-culture. To do this we evaluated whether compounds sharing MOA target annotations return similar representations by computing a running MOA enrichment score (es) similar to Moshkov et al.[18]. This statistical analysis evaluates MOA overrepresentation based on correlation to an input morphological fingerprint vector (Fig. 1d). We selected three main drug classes and targets from the compound library (Supplementary Table 3): kinase inhibitors (CDK, EGFR, MEK1/2, PI3K, VEGFR), conventional chemotherapy (mitotic, topoisomerase inhibitors), and differentiating/epigenetic modifiers (PARP, HDAC, BET). Each of these groups contains multiple compounds with the same MOA and assesses distinct biological mechanisms (Supplementary Fig. 6). For all concentrations of all inhibitors in the selected categories, we computed the number of times running MOA es achieved statistical significance ($p < 0.05$) out of the total number of profiles (corresponding to wells) for the particular MOA as a percentage. High values indicate that the feature representations are similar for all compounds within the MOA, while low values suggest that the compounds or concentrations within the MOA do not elicit a consistent morphological response. Since our library contained concentrations spanning a 10,000-fold range, we expect that there may be drug concentrations where cell death (high concentrations) or no effect (low concentrations) is observed. Notably, the O3B combination stood out in the analysis, where the EfficientNet pre-trained NN outperformed CP in the MOA matching task for 9 out of 10 selected MOAs (Fig. 3a, b, Table 2). This is in sharp contrast with other co-culture combinations where CP features generally outperformed the NN features in our matching task (Table 2, Supplementary Fig. 7). Furthermore, while CP performed similarly across all co-cultures (mean: 62.6%, max: 66.2% for KB, min: 57.3% for O8W), there was a larger variation for the NN (mean: 59.8%, max: 74.1% for O3B, min: 52.6% for O8W). Additionally, MobileNetV2 and ResNet50 models exhibited the same trend, where O3B stood out in the NN-derived embeddings (O3B MobileNetV2 mean: 72.2%, ResNet50 mean: 71.6%), while CP consistently had a higher average MOA recovery than all three NN models (Supplementary Table 4, Supplementary Fig. 7). To assess whether our weak auxiliary matching task was influenced by the inclusion of the query compound's remaining concentrations in the feature vector, we recalculated the es after excluding the wells corresponding to the query compound and its remaining four concentrations. This analysis

## Table 1 | Cancer cell lines, fibroblast cell lines, and co-culture combinations used in this study

| Cancer cell line | Fibroblast cell line | Co-culture combination |
|---|---|---|
| Kuramochi | BjhTERT | KB |
| Kuramochi | WI38 | KW |
| OvCar3 | BjhTERT | O3B |
| OvCar8 | WI38 | O8W |
| MH | BjhTERT | MHB |

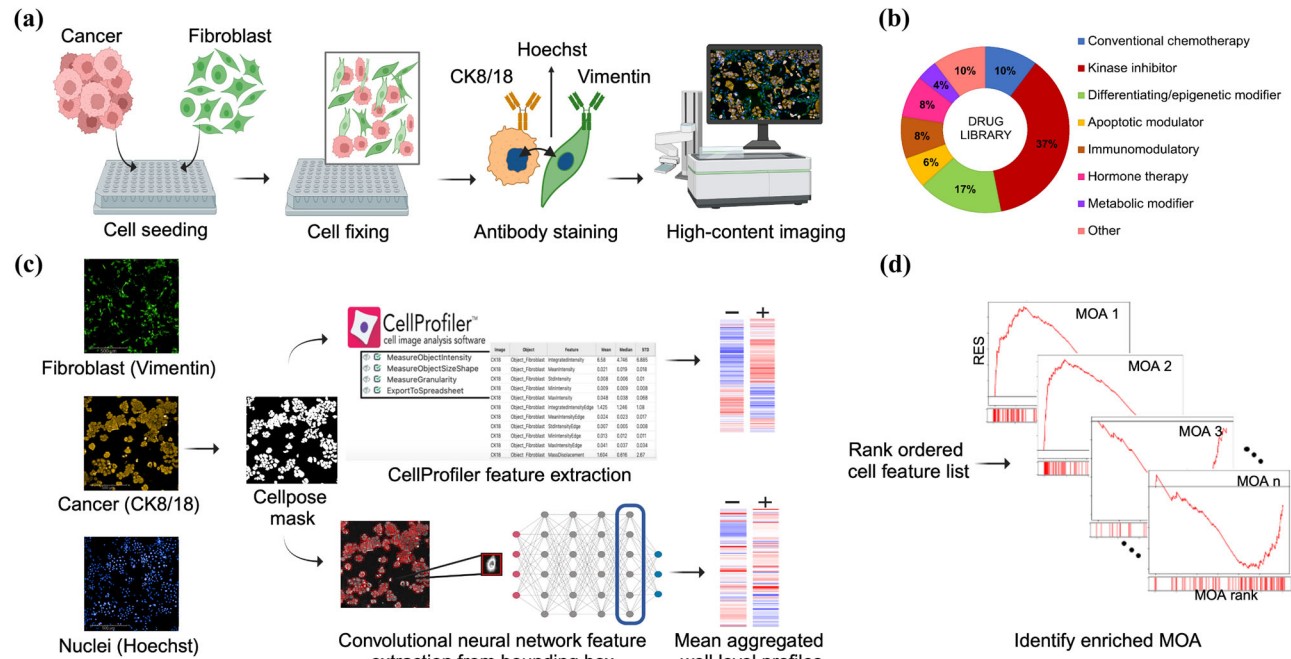

**Fig. 1 | A co-culture dataset to study feature representation learning methods.**
**a** Schematic overview of the generation of the imaging dataset. Antibody staining: CK8/18 (cancer cells), Vimentin (fibroblasts), and Hoechst (nuclei). **b** An overview of drug classes of the 528 drugs from the FIMM Oncology Drug Repurposing Library used in this study. **c** Illustration of the pipeline for CellProfiler and Neural Network to extract morphological single-cell features using cancer cell object binary mask generated by Cellpose. CellProfiler features are directly extracted using cell region from the Cellpose mask. For the Neural Network, bounding boxes (in red) are created around each segmented cancer cell across all three channels (CK8/18,

Vimentin, and Hoechst) and used as input to extract features from an embedding layer of the network. Subsequently, the single-cell features are mean-aggregated to obtain well-level profiles and normalized with respect to negative controls using median and median absolute deviation from the negative controls (DMSO). **d** Overview of the MOA enrichment analysis, an adaptation of gene enrichment analysis which analyzes rank ordered drug list (annotated with the MOA) of features and calculates *p*-values for each MOA category to identify overrepresented MOAs. (Created in BioRender. https://BioRender.com/u03s623).

revealed a moderate decrease in *es* across all methods, indicating that the query compound is not only aligning with its own features but also successfully matching phenotypes of inhibitors within the same class (Supplementary Table 5).

To further investigate this variation, we considered specific differences between the two feature extraction strategies. As stated above, while our NN extracts features from bounding boxes of fixed size with $50 \times 50$ px as a region of interest (Fig. 4), a single-cell object processing method was used in CP. This restricts feature selection to the identified individual cell area. Based on our findings above that each CCL had a distinct morphological profile, we wanted to understand if there were specific morphological factors that account for the observed results. For example, the fixed-size bounding box strategy may not capture enough of the cell region to compute relevant features or may include significant interference from neighboring cells depending on the co-culture growth pattern. To further explore this difference, we examined components of cell morphology, including cell area and cell eccentricity, retrieved from CP (Fig. 3c) using single-cell cancer objects from untreated wells (DMSO).

From this evaluation, we found that MH has the smallest cell area, followed by O8, while *K* in the KW co-culture has the largest cell area. On the other hand, the eccentricity values vary from 0 to 1 and define the roundness of an object, with 0 denoting a perfectly round object and 1 denoting a line segment. Although none of the eccentricity values are close to 0, O3B's lower value suggests that the cell objects may be more rounded than in KW, where the objects appear to have an elongated structure (Fig. 2a, Supplementary Fig. 1) and therefore have a higher eccentricity value. Visual inspection also showed that the nuclei in MHB and KB images contain packed cell colonies (Supplementary Fig. 1). These observations suggest that innate cellular morphologies impact the ability of the NN to extract relevant features using the fixed bounding box.

To investigate whether neighboring cells were influencing feature extraction, we plotted activation maps to visualize the focus of the network (Supplementary Fig. 8a). These maps revealed that surrounding cells exhibited high-intensity activation, indicating that the features extracted by the network were influenced by neighboring cells. While this contextual information may be valuable, it could potentially confound morphological profile changes in individual cancer cell objects. Therefore, we isolated and centered individual cancer cells from neighboring cells and the background by retaining only the pixels within the bounding box corresponding to the Cellpose mask (Fig. 4b). Additionally, the bounding boxes were expanded to $90 \times 90$ pixels to encompass a larger portion of the cell area, based on our calculations of cell area from CellProfiler (Fig. 3c). UMAP dimensionality reduction (Supplementary Fig. 9a) of these masked bounding box features from negative control (DMSO) wells of the cancer cell population again revealed five distinct clusters, each corresponding to specific co-culture conditions (PCA plots in Supplementary Fig. 9b).

Further, all cell line combinations showed a modest improvement when MOA es were computed using masked bounding boxes (mean: 61.0% masked vs 59.8% unmasked) for EfficientNet. The activation maps from the masked bounding boxes showed that the NN concentrated more effectively on the region of interest (Supplementary Fig. 8b), specifically the single-cell object, reducing the influence of the surrounding context and making the comparison between CP and NN more relevant. Notably, we found that MOA es in KB was significantly improved using EfficientNet (mean 67.9% masked vs 56.6% unmasked) (Table 2, Fig. 4c). Overall, the representations from NN feature extraction based on masked bounding boxes using EfficientNet perform similar to CP (mean 62.6% CP, 61.0% NN). Interestingly, NN outperforms CP for KB (mean: 67.9% vs 66.2%) and O3B co-cultures (mean: 63.5% vs 62.1%). Higher es means that more compounds or concentrations are enriched for a specific class, suggesting that for these two

**Fig. 2 | Morphological profiling reveals inherent differences between cell lines and co-cultures.**
**a** Representative images of four ovarian CCLs co-cultured with two fibroblast cell lines (KB, KW, MHB, O3B, O8W) with Hoechst (Nuclei) in blue, CK8/18 (epithelial cells: cancer) in red and Vimentin (stromal cells: fibroblast) in green. UMAP of morphological profiles derived from **b** CellProfiler and **c** pre-trained baseline EfficientNetB0 using 12 DMSO wells per plate across 40 plates for each assay.

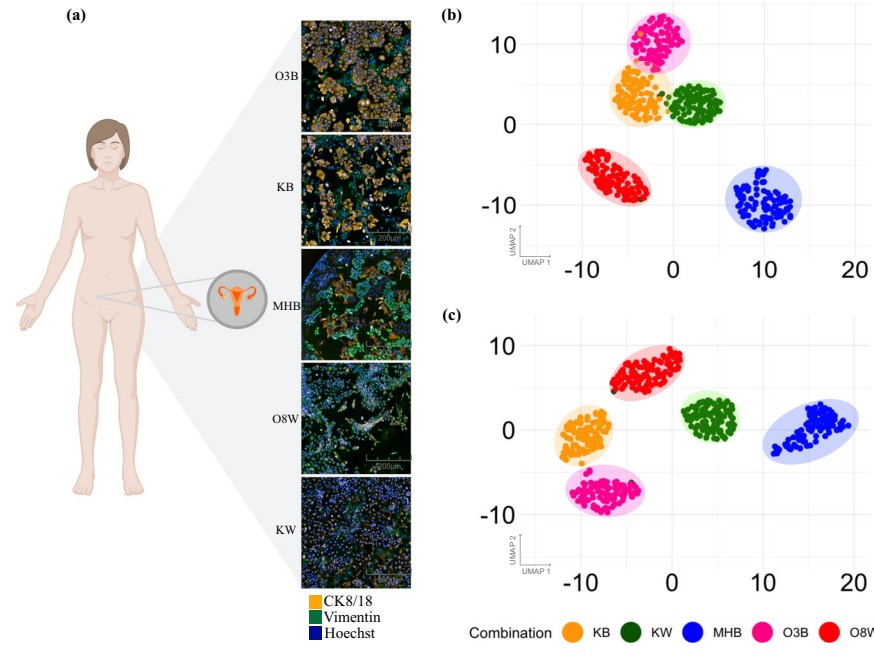

CCLs, the NN based on masked bounding boxes can extract more nuanced response patterns. To understand the dose-dependent trend on es, we computed es separately for the lower and higher concentrations of each drug (Supplementary Table 6). Lower concentrations tend to induce subtle or negligible morphological changes, resulting in weaker es across all methods. Unmasked NN features often struggle with weak patterns due to surrounding cells, whereas masking improves focus but cannot fully overcome the limited response. At higher concentrations, the more pronounced morphological effects lead to stronger enrichment, with both CP and masked NN features effectively capturing these robust responses. Combined, these findings underline the importance of careful experimental design for all methods to ensure the right drug concentrations, as well as robust data preprocessing techniques, to achieve a more accurate and meaningful interpretation of drug-induced morphological alterations in the co-culture setting.

### Comparison of MOA es in co-cultures reveals variation across drug targets and cell lines

Among the MOAs examined, a persistent observation emerged: VEGFR es were consistently low, regardless of whether features were extracted using NN (mean with masked bounding box: 47.6%) or CP (mean: 37.1%) (Table 2). Low es values can arise from inconsistent morphological profiles between small molecules, lack of morphological changes due to treatment, or lack of response to treatment (Supplementary Table 6). Interestingly, the low response to VEGFR inhibition seemed to extend beyond morphological space. We examined the Drug Sensitivity Score (DSS)[35], a metric encompassing drug response data, readouts in co-culture models (Fig. 3d) from cancer cell count (Supplementary Fig. 10), and CellTiter-Glo (CTG) readouts in cancer monoculture models (Fig. 3e). In both cases, the findings consistently indicated low sensitivity to VEGFR, suggesting a lack of response to VEGFR inhibitors. Thus, the low VEGFR es likely reflects a lack of response to VEGFR inhibitors in our experimental setup and all three masked NN's outperform CellProfiler for this class of compounds. Interestingly, however, the lower two concentrations give a higher es across all feature extraction methods, suggesting that the phenotype is more consistent for these treatments. This is the only MOA where this trend is robustly observed (Supplementary Table 6). For the EGFR compound class, DSS directly correlates with *es*. MH, the most sensitive cell line, has the highest es, and the least sensitive cell line O8 has the lowest es. (Table 2, Supplementary Table 4, Supplementary Figs. 7 and 11).

In contrast, there are instances where DSS is low, suggesting minimal drug response, but the MOA es was high. This suggests that these classes of drugs may not significantly affect traditional cell viability readouts, but do induce consistent alterations in cell morphology. For example, although PARP inhibitors have low DSS across all cell lines (Fig. 3d, e), this class is enriched in >75% of the wells in 4/5 co-cultures using EfficientNetB0 with masked bounding boxes (Table 2). This could be an indication that the drugs have specific effects on cellular structures, signaling pathways, or other morphological features without significantly decreasing cell viability measured based on changes in ATP levels. In MHB and KW, we observe an interesting contrast where DSS for PI3K is high, indicating the sensitivity of these cell lines to drugs targeting the PI3K pathway. However, the corresponding enrichment for PI3K-associated morphological features is unexpectedly low. This discrepancy suggests that, despite the cells showing high sensitivity to PI3K-targeting drugs, the distinct morphological changes linked to PI3K activity are not as consistent. For both KB and O3B, which have similar DSS as MHB and KW, the es is considerably higher, suggesting that cell line and context influence MOA recovery. This is particularly interesting since K cultured with B or W gives different results depending on the co-culture context. Overall, we demonstrate that the imaging data can recover nuanced cellular responses that are not captured in viability measurements. These findings highlight the importance of evaluating cell lines with appropriate genetic and cellular signaling dependencies along with tumor microenvironments to recover clustering and identify MOA accurately[36–38].

As seen above, directly comparing feature extraction methods for the KB and KW co-cultures enables us to examine how they perform in different cellular contexts, since we are examining the same cancer cells. For CP, the largest discrepancy in es is for the targeted therapies: EGFR, MEK1/2, and PI3K. Again, this suggests that cellular context can significantly influence the ability to recover phenotypic outcomes of drug treatment based on morphological changes[28]. Interestingly, the DSS for these inhibitors is significantly higher for KB than KW (Supplementary Fig. 12). For the masked NN using EfficentNetB0 or MobileNetV2, these differences are distinctly less pronounced. Instead, there are larger differences in es for BET inhibitors (Table 2). These inhibitors are slightly more effective in KW than KB (Supplementary Fig. 12). The unmasked baseline NNs have similar trends to CP, suggesting here, that nearby fibroblasts are impacting the MOA matching in the KW setting and that the overall lower drug response observed in KW confounds this further. These findings indicate that cellular

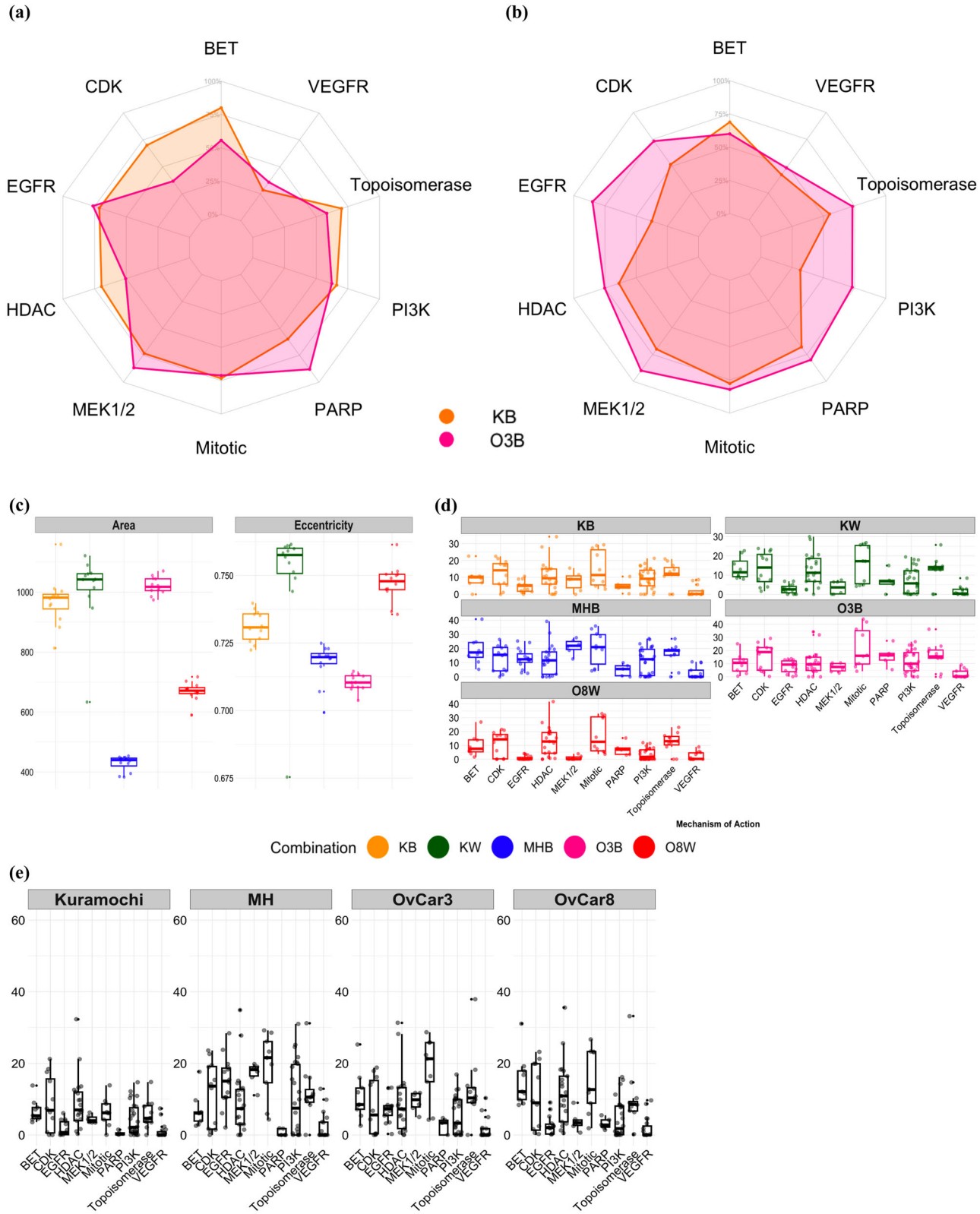

**Fig. 3 | MOA enrichment analysis reveals strengths and limitations of CP and NN-based feature extraction, quantitative analysis of cellular interactions, and morphological diversity.** Enrichment radar plots showing the percentage of significantly enriched ($p < 0.05$) morphological feature profiles extracted from **a** CellProfiler and **b** pre-trained baseline EfficientNetB0 model for O3B and KB co-cultured assays. **c** Box plots illustrate the distribution of affected morphological features, area, and eccentricity, within cancer cells as regions of interest for each CCL treated with DMSO. **d** Drug sensitivity score (DSS) from co-culture assays depict varied drug sensitivity patterns across cell lines, reflecting the influence of fibroblasts on cancer cell growth and highlighting differential responses to drugs with distinct MOAs. **e** Box plots representing cancer cell monoculture DSS based on CTG viability assay. All the box plots have black lines representing the median of the data.

**Table 2 | Running enrichment score for 10 selected modes of actions from pre-trained baseline EfficientNetB0 model**

| Mechanism of Action (MOA) | es from CellProfiler (CP) features | | | | | average CP per target | es from EfficientNetB0 features with unmasked bounding box | | | | | average NN per target | es from EfficientNetB0 features with masked bounding box | | | | | Average NN per target (masked) |
|---|---|---|---|---|---|---|---|---|---|---|---|---|---|---|---|---|---|---|
| | KB | KW | MHB | O3B | O8W | | KB | KW | MHB | O3B | O8W | | KB | KW | MHB | O3B | O8W | |
| BET | 80.0 | 86.7 | 75.6 | 55.6 | 75.6 | 74.7 | 68.9 | 77.8 | 71.1 | 60.0 | 64.4 | 68.4 | 88.9 | 48.9 | 66.7 | 51.1 | 55.6 | 62.2 |
| CDK | 70.0 | 73.3 | 26.7 | 36.7 | 41.7 | 49.7 | 51.7 | 35.0 | 35.0 | 73.3 | 6.7 | 40.3 | 61.7 | 68.3 | 23.3 | 23.3 | 43.3 | 44.0 |
| EGFR | 71.3 | 48.8 | 81.3 | 76.3 | 13.8 | 58.3 | 37.5 | 41.3 | 85.0 | 85.0 | 16.3 | 53.0 | 68.8 | 60.0 | 78.8 | 82.5 | 27.5 | 63.5 |
| HDAC | 69.5 | 77.1 | 72.1 | 50.5 | 74.3 | 68.7 | 63.8 | 76.2 | 51.0 | 75.2 | 68.6 | 67.0 | 54.3 | 61.0 | 55.8 | 53.3 | 63.8 | 57.6 |
| MEK1/2 | 73.3 | 33.3 | 93.3 | 86.7 | 76.7 | 72.7 | 70.0 | 23.3 | 96.7 | 90.0 | 80.0 | 72.0 | 66.7 | 53.3 | 76.7 | 83.3 | 76.7 | 71.3 |
| Mitotic | 73.3 | 91.1 | 73.3 | 71.1 | 75.6 | 76.9 | 77.8 | 66.7 | 66.7 | 82.2 | 68.9 | 72.4 | 80.0 | 73.3 | 57.8 | 62.2 | 66.7 | 68.0 |
| PARP | 60.0 | 76.0 | 52.0 | 88.0 | 68.0 | 68.8 | 68.0 | 72.0 | 68.0 | 80.0 | 72.0 | 72.0 | 64.0 | 84.0 | 88.0 | 88.0 | 76.0 | 80.0 |
| PI3K | 66.2 | 25.4 | 38.8 | 62.3 | 34.6 | 45.4 | 31.5 | 53.8 | 27.1 | 73.1 | 56.9 | 48.5 | 62.3 | 42.3 | 33.3 | 73.8 | 34.6 | 49.3 |
| Topoisomerase | 70.0 | 90.0 | 75.0 | 58.3 | 76.7 | 74.0 | 55.0 | 80.0 | 58.3 | 73.3 | 78.3 | 69.0 | 68.3 | 70.0 | 61.7 | 61.7 | 71.7 | 66.7 |
| VEGFR | 28.4 | 32.6 | 52.6 | 35.8 | 35.8 | 37.1 | 42.1 | 36.8 | 33.7 | 48.4 | 13.7 | 34.9 | 64.2 | 32.6 | 37.9 | 55.8 | 47.4 | 47.6 |
| Average | 66.2 | 63.4 | 64.1 | 62.1 | 57.3 | 62.6 | 56.6 | 56.3 | 59.3 | 74.1 | 52.6 | 59.8 | 67.9 | 59.4 | 58.0 | 63.5 | 56.3 | 61.0 |
| | | | | | es > 80 | 7.0 | | | | | es > 80 | 5.0 | | | | | es > 80 | 6.0 |
| | | | | | es > 75 | 16.0 | | | | | es > 75 | 13.0 | | | | | es > 75 | 11.0 |
| | | | | | es > 70 | 25.0 | | | | | es > 70 | 19.0 | | | | | es > 70 | 14.0 |
| | | | | | es > 65 | 30.0 | | | | | es > 65 | 27.0 | | | | | es > 65 | 21.0 |
| | | | | | es > 60 | 31.0 | | | | | es > 60 | 29.0 | | | | | es > 60 | 30.0 |

context can have a large impact on the extent of MOA recovery and that the feature extraction methods are affected to different extents.

### Enhanced training dataset through combined co-culture wells

Cell lines exhibit diverse morphological characteristics, including variations in shape, size, and structure, posing a challenge for pre-trained baseline NN trained on generic image datasets. As a result, these networks may not fully capture the nuances of cell morphology. To address this, we fine-tuned the ImageNet pre-trained EfficientNetB0 model for two tasks: (1) MOA multiclass classification, where single-cell objects are classified into their respective MOA, and (2) binary classification to distinguish between treated vs non-treated cells. This additional training was motivated by the observation that the pre-trained baseline EfficientNetB0 masked model showed improvements across all co-culture combinations except MHB, prompting us to explore whether further training could enhance performance. To account for potential dose-dependent variability, we used the middle three drug concentrations (Fig. 5a), ensuring the network was exposed to a broader range of cellular morphologies and drug effects. The results showed that the MOA-trained model performed similarly to CellProfiler features (mean: 64.1% vs. 62.6%) (Fig. 5b), while the binary-trained model performed slightly worse overall (mean: 54.3%) (Fig. 5c) (Supplementary Table 7). The binary classification task may not be optimal for this data, as some drugs may fail to induce noticeable morphological changes at the tested concentrations. This could lead to misclassification, where treated cells are inaccurately labeled as non-treated due to the absence of detectable morphological alterations. In contrast, training the network for MOA classification is a more clearly defined task, as it directly targets drug-induced morphological patterns. Additionally, as previously discussed, es is not expected to be very high due to dose-dependent effects; at lower concentrations, drugs may induce minimal or no morphological changes, making it challenging to capture such subtle responses.

As ResNet50 showed lower es overall among the three selected models (mean 57.9% for masked), with the same approach of taking the middle three concentrations for training, we further fine-tuned ResNet50 using pooled wells from all co-culture combinations, focusing on the 10 previously selected MOAs. To evaluate whether the pre-trained baseline ResNet50 and the fine-tuned model could still distinguish negative and positive control populations, we plotted UMAPs (Supplementary Figs. 13 and 14). The results demonstrated that both models extracted features capable of separating negative and positive control wells effectively. The fine-tuned model has significant improvement in KB (Supplementary Table 8). For this cell line, the refined model gave greater es values than the pre-trained baseline ResNet50 in the matching task for 5 of the 10 MOAs (es >75%). The model improves performance for certain MOAs, such as BET (KB: 82.2% vs. 44.4%), EGFR (KW: 51.3% vs 48.8%, MHB: 78.8%), and Mitotic, particularly in co-cultures O3B and O8W. The baseline model outperforms the fine-tuned model in O3B (69.2% vs. 67.6%) and O8W (57.3% vs. 53.6%). Despite variations across co-cultures, the fine-tuned model achieves a slightly higher overall average performance across all co-cultures and MOAs (61.9% fine-tuned vs. 57.9% baseline). This suggests that, on average, fine-tuning provides better adaptability for most MOAs, even though the baseline model remains competitive in certain contexts.

All the es calculations for each co-culture combination, broken by drug and concentrations are reported in the Supplementary Data file.

### Discussion

This paper examines and compares feature extraction using CP and three pre-trained baseline NNs. In addition, we evaluate (1) NNs with complete bounding box area, (2) NNs with masked bounding box area, and (3) fine-tuned NNs utilizing masked bounding box area from the middle three concentration wells. These methods are investigated for their efficacy in recovering similar feature representations across 10 MOA classes in 2D co-culture assay imaging experiments. The weak auxiliary matching task was our MOA enrichment analysis framework. We used this setup to study five different combinations of cancer and fibroblast cells, each treated with 528

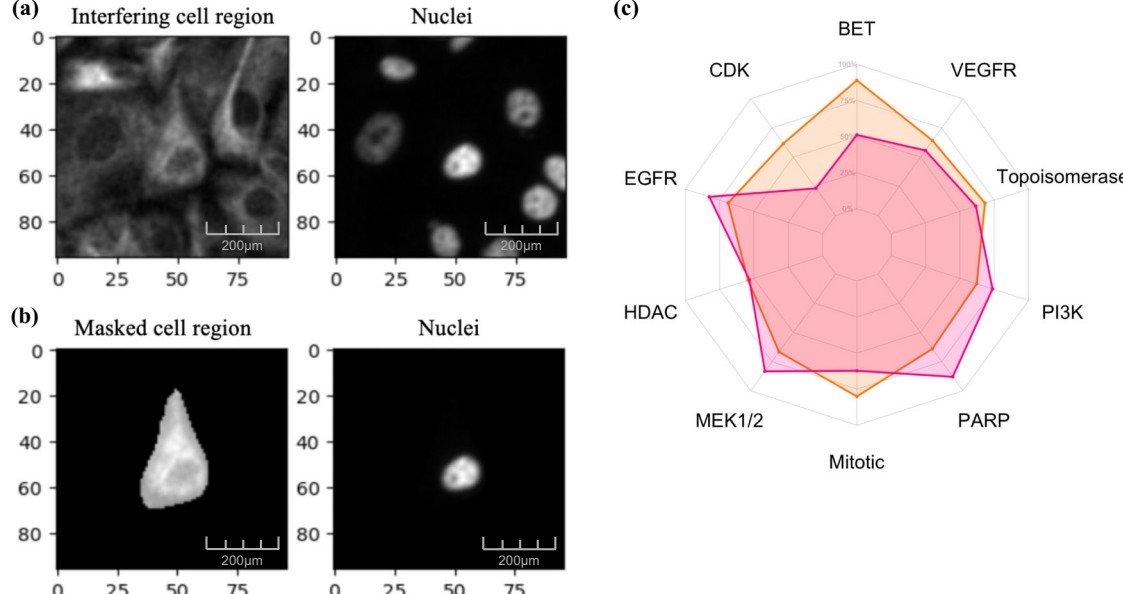

**Fig. 4 | Region of interest masking improves NN-based morphological feature extraction for certain cell lines. a** Unmasked bounding box encompassing neighboring cells from the cancer channel (left) alongside its corresponding nuclei channel image (right). **b** Masked image with only a single cell as the region of interest (left), paired with its corresponding nuclei (right). **c** Presents enrichment radar plots obtained from morphological features extracted from baseline EfficientNetB0 within the masked bounding box region for O3B (pink) and KB (orange) co-cultured assays.

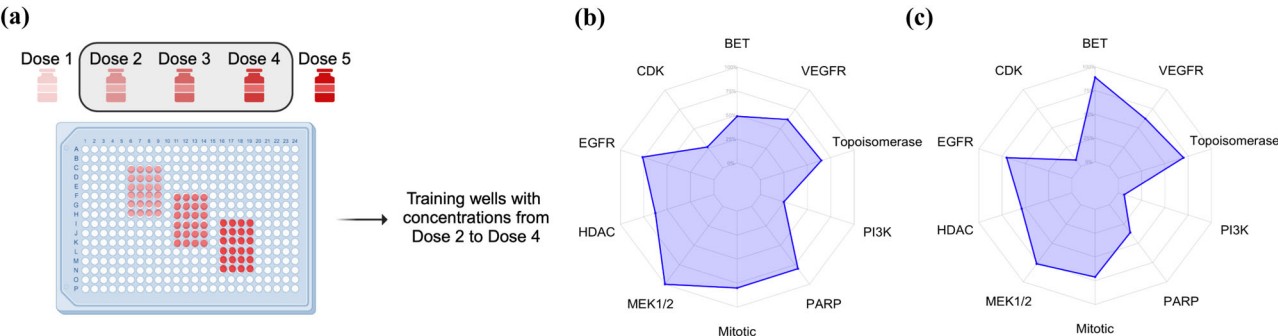

**Fig. 5 | Training using heterogeneous cell lines and drug concentrations improves MOA recovery in specific cell lines and targets. a** Random fractions of cells from all 8 plates in the MHB dataset were utilized for training, focusing solely on doses 2–4, representing the middle three concentrations of drugs. Enrichment radar plots showing the percentage of enriched features **b** when trained for multiclass classification tasks. **c** when trained for the binary classification task of classifying cells as treated vs non-treated. (Fig. 5(a) Created in BioRender. https://BioRender.com/u03s623).

drugs at five concentrations. The dataset generated in this study is made available to the research community, facilitating future efforts in single-cell analysis and drug screening studies.

Our study found that: (1) Including contextual information (like neighboring cells) might also add noise from nearby areas. (2) Restricting information to cell object area yields more comparable MOA es scores between NN and CP (Supplementary Tables 2 and 4). However, this approach might lose some contextual clues, affecting model performance depending on the balance between precision and context. (3) Pre-trained NNs demonstrate superior performance when applied to cell lines whose morphological characteristics closely align with the general low-level features learned by the model. However, for other cell lines with unique characteristics, the pre-trained network might not generalize. (4) The sensitivity or resistance of cell lines to drugs can affect how well certain biological processes are represented in the data and, therefore, recovered in the analysis. This confounds our ability to predict MOA by measuring phenotypic effects in a limited number of cell line models. Since the es calculation included all compounds and their concentrations, we checked whether excluding the query compound and its four remaining

concentrations would affect the results. This analysis revealed a moderate decrease in ES across all methods, indicating that the query compound aligns not only with its own features but also successfully matches phenotypes of inhibitors within the same class (Supplementary Table 5).

Our weakly supervised learning approach aims to learn unbiased features of cellular morphology, which can be useful for various applications in cell biology. To address the underperformance in es observed in MHB co-cultures (Supplementary Table 2), we trained the EfficientNetB0 model with two different strategies. First, with a multiclass classification task for MOA and the other with a binary classification task to distinguish treated vs. not treated cells. The results revealed that the multiclass classification approach boosted performance significantly, while the binary classification task struggled to extract features relevant to MOA classes (Supplementary Table 7). This suggests that the binary classification task was not well-suited for this dataset and instead, a more supervised selection of wells for the training could improve the overall performance of the approach.

Additionally, we explored training ResNet50 on pooled samples from all co-cultures, focusing on the middle three compound concentrations. We opted to train ResNet50 primarily because of its widespread usage and

effective feature extraction[39–41]. While other methods like DeepProfiler are available, which utilize weakly supervised learning approaches[18], we did not find them compatible with our dataset. Despite DeepProfiler's capability to handle more complex tasks, such as five-channel data, our experiment did not require such complexity. This study, conducted at 10x magnification, suggests that images captured at higher magnification may offer better morphological representations of single cells[42,43]. While employing antibody staining in this study provides a targeted approach, leveraging fluorescent images for MOA prediction could involve integrating antibody staining with Cell Painting[13] or utilizing Cell Painting alone. Given its untargeted approach and ability to stain different cellular compartments, Cell Painting may enhance the prediction of morphological features by providing more comprehensive cellular information[13,44].

Future research could explore the concept of a "multi-culture" assay, incorporating more than 2 CCLs along with fibroblasts to encompass a wider range of mutation combinations. This approach would replicate the in-vivo complexities of the disease and enable the prediction of drug effects more accurately. A data fusion approach could be explored by integrating compound structural information with fluorescent images[32,42,45]. By combining these diverse data modalities, we can potentially gain deeper insights into drug-induced phenotypic effects leading to more robust and comprehensive predictions regarding the MOA of various compounds, thus advancing drug discovery and development efforts. Overall, this study is a first step towards understanding how different fibroblast types and drug treatments affect the cancer cell population using high-throughput imaging experiments and modern deep learning techniques.

## Methods

### Cell culture

Ovarian CCLs Kuramochi, OVCAR8, OVCAR3, and MH[46] were cultured in RPMI medium supplemented with 10% FBS, 1× antibiotics (streptomycin/penicillin), and 1× L-glutamine. The fibroblast cell line BjhTERT was maintained in DMEM supplemented with 14.5% FBS, 16.5% Medium 199, and 1% antibiotics, while the lung fibroblast WI38 – MEM/EBSS medium supplemented with 10% FBS, 1% antibiotics, 1% non-essential amino acids. WI38 cell line was purchased from VWR, and the remaining cell lines were donated from the Precision Systems Medicine research group in FIMM. All cell lines were confirmed by short-tandem repeat sequencing and were tested for mycoplasma. Cells were maintained in a 5% $CO_2$-humidified incubator at 37 °C and passaged 2 times a week when confluency reached ~70%.

### Immunofluorescence staining and image acquisition

A library of 528 oncology compounds was used to assess phenotypic outputs (Supplementary Table 1)[31]. Assay-ready 384-well plates (Revvity, 6007688) were prepared using an acoustic liquid dispensing system (Echo550, BeckmanCoulter). Cells were seeded into the pre-spotted drug plates and treated for 72 h. Then cells were fixed with 4% paraformaldehyde (PFA, ThermoFisher Scientific) for 10 min, followed by 10 min permeabilization with 0.3% Triton-100X and 1 h blocking with 3% bovine serum albumin (BSA, VWR). Plates were incubated overnight at 4 °C with a primary antibody cocktail: CK8/18 (M3652, Agilent) and vimentin (MA5-11883, Invitrogen) at a final dilution of 1:300 each. The next day secondary antibodies Donkey anti-rabbit (A10042 and A21206, Invitrogen) and Donkey anti-mouse (A21202 and A10037, Invitrogen) were added, and plates were incubated for 1 h at room temperature. Subsequently, cells were stained with Hoechst 33342 (Invitrogen, 1:10,000 dilution) for 5 min at room temperature. Washes were performed between each step using the EL405 (Biotek) liquid handling system. Stained plates were stored at 4 °C and imaged within 1 week after staining. Images were acquired using a high-content imager (OperaPhenix, Revvity) with a 10X objective (numerical aperture 0.3). Four fields of view (FOV) were acquired with a 5% overlap to ensure comprehensive coverage. Additionally, a 2 × 2 binning was applied during image acquisition to enhance the signal-to-noise ratio and improve image quality.

### Drug sensitivity testing of monocultures

The same assay-ready drug plates as described above were used to treat cancer and fibroblast cells in monoculture. After 72 h of drug treatment, cell viability was assessed using the CellTiter-Glo (Promega) assay, and the luminescence signal was measured using a multimodal plate reader (EnSight, PerkinElmer). The acquired data was used to calculate drug sensitivity scores (DSS) using web-based Breeze software (FIMM)[35,47].

### Image preprocessing

After exporting the raw images, preprocessing was divided into two steps: (1) merging the four FOVs, and (2) illumination correction. The merging of the FOV images per well involved several steps. Initially, each FOV image was converted into a NumPy array using the PIL library. Then, the arrays representing the individual FOV images were horizontally stacked in pairs, considering the overlap at the edges. This produced two horizontally merged arrays. Subsequently, these merged arrays were vertically stacked to generate the final well-level image. This concatenation process ensured seamless integration of the FOV images while accounting for edge overlap between adjacent FOVs. This process yielded three images per well, each with dimensions of 2130 × 2130 pixels, per well. The next step involved addressing the challenges of microscopy imaging, and variations in the light source which results in uneven illumination across the wells, leading to brighter or darker regions near the edge of the plates. This can cause challenges in image analysis due to inconsistent signal intensities. Additionally, staining protocols may result in background fluorescence or noise. To address these issues, the CP modules CorrectIlluminationCalculate and CorrectIlluminationApply were employed on each merged channel image to normalize the intensity signal across fluorescent channels, ensuring uniform illumination and reducing the impact of background signal interference from other channel stains.

### Segmentation and bounding box creation

In this project, our focus was primarily on analyzing the morphological variations within the cancer cell population, despite the dataset comprising a co-culture of both cancer and fibroblasts. To accomplish this, we employed the Cellpose cellular segmentation algorithm on illumination-corrected, merged cancer channel images to generate cell masks (Supplementary Fig. 3). The Cellpose masks were directly used to extract features from the CP pipeline, for NN these masks were utilized to create bounding boxes for each cell across all the 3 channels i.e. cancer, fibroblast, and nuclei using a function from the skimage.measure module, which is a part of the scikit-image library in Python. The function returns a list of properties for a specified region, such as area, perimeter, centroid, and bounding box dimensions. We used the centroid coordinates of each cell object to create bounding boxes, and subsequently employed array manipulation to extract the region of interest, yielding a box size of 50 × 50 pixels. Cells located at the edges, which did not yield a complete 50 × 50 pixel box, were excluded from the analysis to ensure the integrity of the data. However, this approach resulted in the inclusion of neighboring cells within the box region, which was suboptimal, as it led to the analysis of neighboring interfering cells alongside the target cell. Consequently, accurate comparison with CP features became challenging, and CP performed better in this context. To address this issue and ensure the analysis and comparison of the same kind of image data, we refined the bounding box regions using masks from Cellpose. Additionally, we increased the size of the bounding box to 90 × 90 pixels, thereby incorporating a larger portion of the cell object while excluding neighboring cells. This adjustment aimed to improve the accuracy of the analysis and facilitate a more meaningful comparison between CP and NN features.

### Extraction of features from CellProfiler

We retrieved 236 features from the three-channel fluorescence images. For additional information on the CP feature extraction process, see the CP pipeline under the "Data availability" section.

## Extraction of features from pre-trained baseline neural networks

Before extracting features from the NN, we conducted additional pre-processing of the image data to improve contrast and standardize pixel intensity ranges. This involved normalizing each channel separately. For each channel image, we divided the pixel intensities by the 99th percentile value of that specific channel and then multiplied the result by 255 to scale the pixel intensities. Any values exceeding 255 were clipped to 255, ensuring the pixel intensities remained within the range of [0,255]. We chose to normalize based on the 99th percentile to enhance contrast and use the dynamic range of pixel intensities. This process was implemented using the NumPy library for efficient array operations and percentile calculations, along with Python scripts that automated the normalization across the entire dataset. Subsequently, we extracted features from normalized images using three pre-trained baseline convolutional neural network architectures: EfficientNetB0, MobileNetV2, and ResNet50. EfficientNetB0 and Mobile-NetV2 were specifically chosen for their smaller architectures and fewer parameters, making them well-suited for handling our relatively small input image bounding box sizes of $50 \times 50$ px and $90 \times 90$ px. These networks provide an efficient alternative for feature extraction without compromising performance on small-scale data. ResNet50, a deeper network with a residual architecture, was also included to compare the performance of larger networks against smaller ones. All models utilized transfer learning, leveraging weights pre-trained on the ImageNet dataset to extract meaningful features from the new dataset[24,39,48,49]. Feature extraction was implemented using the Keras API within the TensorFlow library, enabling efficient integration of these pre-trained models. To obtain features from the cancer cell population, bounding boxes for each cell in all three channels were transformed into a NumPy array and were used as input to the network. The features were then extracted from the embedding layer following Global Average Pooling. See the "Data availability" section for Python scripts.

## Model training

We trained the EfficientNetB0 model specifically for MHB co-culture assays using two distinct approaches. In the first approach, we utilized samples from the middle three concentrations of drugs belonging to selected MOA categories. To reduce computational complexity, single-cell bounding boxes were randomly selected within each sample. These single-cell data were then split into training, test, and validation sets using stratified splitting based on MOA, ensuring balanced representation. To address data imbalance, class weights were applied during training. The model employed a data generator and augmentation techniques, including width shift, height shift, and shear transformations. A custom loss function incorporating class weights was utilized to further mitigate class imbalance. Starting with ImageNet-pre-trained weights, the model was fine-tuned with a new classification layer to classify single cells based on their MOA.

In the second approach to training the MHB model, we incorporated a weak auxiliary task by classifying cells as treated versus untreated. This binary classification task included MOA samples from the middle three drug doses, along with negative controls (DMSO). This auxiliary task aimed to enhance the model's performance by leveraging the binary classification signal while maintaining the focus on the primary MOA classification.

Additionally, we trained a ResNet50 model using pooled samples from all co-culture assay combinations. For this model, we focused on a binary classification task of distinguishing treated versus untreated cells. Similar to the MHB models, samples from the middle three drug concentrations were used, ensuring consistency across datasets. By pooling data from all co-culture assays, this approach enabled a broader assessment of the binary classification task.

## Single-cell feature aggregation and normalization

For downstream analysis of single-cell features, which can be computationally intensive and time-consuming to handle, we aggregated single-cell features to obtain well-level profiles. We computed the mean value for each feature across all cells within a well, producing a single vector representation for each well. To ensure robustness in our results, we further normalized the aggregated features using the median and median absolute deviation of the negative control samples (DMSO). According to Caicedo et al. [2], this normalization procedure within plates aids in attenuating batch effects.

## Mechanism of action enrichment analysis

The MOA enrichment analysis employs a similar algorithm to Gene Set Enrichment Analysis (GSEA)[50] but operates on sets of MOAs to identify overrepresented MOAs within the ranked feature list. The es and corresponding $p$-value for each MOA category were computed to assess the degree of enrichment among wells exhibiting high correlation coefficients for morphological features. The analysis involves several steps to ensure robustness and accuracy in identifying significant MOA categories.

Initially, the dataset was prepared to include MOA annotations and correlation coefficients derived from morphological features. For each MOA category, the es was calculated by sorting correlation values in descending order and computing a running sum, where positive increments were applied for matches to the category and negative increments for non-matches. The maximum value of the running sum was recorded as the es. To determine the significance of the calculated es, MOA category labels for each sample (excluding the first sample) were randomly shuffled 1000 times, and es were recalculated for each shuffle. A $p$-value was computed as the proportion of shuffled es values that exceeded the original es. MOA categories with $p$-values less than 0.05 were considered significantly enriched, indicating non-random associations with highly correlated samples based on features.

In a secondary approach, enrichment scores were recalculated excluding query compounds and their concentrations to minimize potential biases. For each query compound, all other concentrations of the same compound were excluded from the dataset. A filtered correlation matrix was constructed without these entries while retaining all other data. Using this filtered dataset, es was calculated for each MOA category following the same running sum method as described earlier. Statistical significance was assessed by shuffling MOA category labels and re-calculating es in the same manner as the full-data approach. This additional layer of analysis ensured that the observed enrichments were not driven by the presence of specific compound concentrations, providing a more nuanced understanding of MOA associations in the dataset.

## Statistics and reproducibility

$Z'$ scores were calculated for each plate to evaluate the separation between positive (BzCl, $n = 12$) and negative (DMSO, $n = 12$) controls. A $Z'$ score $> 0.5$ indicated adequate separation between the controls. For normalization of features, median and median absolute deviation (MAD) were calculated on per plate basis from the negative control wells as follows:

$$\text{Normalized features} = \frac{\text{feature value} - \text{median of controls}}{\text{MAD of controls}}$$

es was calculated by performing 1000 random shuffles to generate a null distribution. A $p$-value $< 0.05$ was considered significant. The percentage of enrichment for each MOA was computed using the following formula:

$$\text{Percentage enriched} = \frac{\text{Number of MOA enriched}}{\text{Total number of MOA samples across per coculture}} * 100$$

## Data availability

Raw image files for the dataset analyzed in the manuscript can be found here: https://snd.se/sv/catalogue/dataset/2024-175

## Code availability

https://github.com/Functional-Precision-Medicine-Lab/Evaluating-Feature-Extraction-in-Ovarian-Cancer-Cell-Line-Co-Cultures-Using-Deep-Neural-Networks

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

## Acknowledgements
The authors acknowledge Francesco Marabita for the compound annotations. B.S.L. acknowledges grants from the Swedish Research Council (2021-03420), Karolinska Institute Doctoral funding (2020-01096), and Sagen Stiftelse (2022-RSL). O.K. acknowledges the Swedish Research Council (2017-06095) and Knut and Alice Wallenberg Foundation (2015.0291). L.P. acknowledges the Research Council of Finland grant (340273).

## Author contributions
G.G., R.M., and B.S.L. performed cell culture experiments, high-content imaging, and initial quality control. O.S. performed feature extraction from CellProfiler, pre-trained neural networks, and fine-tuned/trained neural networks. O.S. analyzed the morphological features. B.S.L., L.P., and R.T. provided input during result analysis and participated in thorough discussions. O.S. made all the figures. O.S. and B.S.L. wrote the first draft of the manuscript. B.S.L. and R.T. designed the project and B.S.L., R.T., L.P., and O.K. supervised the project. All authors reviewed, edited, and contributed to discussions on the manuscript.

## Funding

## Competing interests
O.K. is a Board member and co-founder of Sartar. Additionally, he serves as an advisor to the Knut and Alice Wallenberg Foundation, the Novo Nordisk Foundation, and Sitra.
