## [Transparent Peer Review file · Communications Biology]

Evaluating Feature Extraction in Ovarian Cancer Cell Line Co-Cultures Using Deep Neural Networks

Corresponding Author: Ms Osheen Sharma

Version 0:

Reviewer comments:

Reviewer #1

(Remarks to the Author)
COMMSBIO-24-4129 review:

In the manuscript 'Evaluating Feature Extraction in Ovarian Cancer Cell Line Co-Cultures Using Deep Neural Networks' by Sharma et al. different classification strategies for co-cultured cancer cell drug assays are studied. In particular, an AB-staining microscopic high-content screen of five different combinations of ovarian cancer cell line and fibroblast co-cultures under five different drug concentrations was used for testing (245K image data set provided). Based on cell nuclei segmentation masking, two different concepts of feature extraction were evaluated using specific scoring derived from gene enrichment of the used drugs (modes of actions): (i) defined morphological features (ii) convolutional feature maps of neural networks (NN).

Without referring to any publications or reading anything here the (AI) research community is well aware of the better performing concept.

Consequently, almost all figures in manuscript show improved scoring with the generic convolutional concept. It is obvious for the reader with a machine learning background that the shape and size of the input information in cell co-cultures is key for the classification performance, which is also found when expanding the input bounding box after segmentation for NN, however not further evaluated in greater detail in this study. Biological findings are limited to general impact of fibroblast co-cultures and drugs to ovarian cancer cells and specifics in drug sensitivities (e.g. VEGFR or PI3K pathway). From 528 compounds only few selected drugs treatments are exemplified in the manuscript. The training/test procedure for the NN is not further explained.

Overall, given the few biological insights depending on neighborhood/drug sensitivities and obvious statements regarding feature extraction and classification this study might not be considered for publication in Nature Communication.

Reviewer #2

(Remarks to the Author)
Brief summary of the manuscript

The manuscript presents the image-based profiling analysis of the data of five combinations of ovarian cancer and fibroblast cell lines. The work does not introduce novel computational methods, but rather analyses the in-house data of interest and shares the experience and observations using an established methodology.

Overall impression of the work

The work presents a carefully done analysis. The writing is mostly clear and the figures are also clear and informative. The work misses a few details which could be clarified in the text and maybe more data is desired which could improve the paper. All those questions and concerns are in the comments.

Specific comments, with recommendations for addressing each comment

1. For such small images (50x50 and 90x90), would it make more sense to use even smaller models, like ResNet 34\18?
2. Masked single-cells vs unmasked seem to be complementary in some cases. For example, PARP AND BET seem to work better for O8W and KW (supp. figures 8,11).
Would it make sense to concat \ pool features from surroundings and the cell?
3. Could you clarify the following: when calculating the enrichment score, was the query treatment in other concentrations excluded as a potential response?
4. Do you think it would make sense to report ES per drug per concentration to see at which concentration different drugs make a visible impact on the cell? Can it be the case that CellProfiler and NN detect drug effects at different concentrations?
5. Lines 230-238: Can it be the case that PI3K-targeting drugs just kill the cells, similarly to other drugs which results in phenotypic inconsistency? Drugs of which target pop up at the top of the list if the PI3K-targeting drug was used as a query for enrichment analysis?
6. Is there a correlation between the distance of consensus treatment profiles from consensus negative control profiles and cancer cell count and DSS?
7. Could you clarify whether the initial (not enhanced) dataset used all concentrations or only one per treatment?
8. Do you think that your work would benefit if you tried some explanation technique (GradCAM)? For instance, with that, you could potentially observe the Rols for the NN if the neighbouring cells are in the crop and what changes if not.
9. I see that you have used trained models. I think you could include a baseline of ImageNet-pretrained network without further training.

Reviewer #3

(Remarks to the Author)

The manuscript titled "Evaluating Feature Extraction in Ovarian Cancer Cell Line Co-Cultures Using Deep Neural Networks" investigates the application of deep neural networks, specifically ResNet50, for feature extraction in a high-content imaging (HCI) setting, focusing on ovarian cancer cell lines co-cultured with fibroblasts. The study contrasts traditional feature extraction methods using CellProfiler (CP) with a neural network (NN)-based approach, particularly in how these methods capture drug-induced morphological changes. A large dataset was generated from five different co-culture combinations treated with a library of 528 oncology drugs to assess the phenotypic effects of these treatments. The primary findings suggest that while both CP and NN capture significant variation in cellular morphology, the NN approach slightly outperforms CP in more complex model systems. The study also underscores the critical role of data preprocessing techniques in ensuring accurate interpretations of drug-induced morphological changes. While the work presented by the authors is commendable, several caveats need to be addressed.

Main Comments:

1. Optimization of the Neural Network (NN) for Feature Extraction: Although using ResNet50 for feature extraction is a standard approach, it may require further optimization in this context. ResNet50 is pre-trained on larger tiles (224x224 pixels), which might not be ideal for smaller images (50x50 or 90x90 pixels). The authors should consider exploring other models better suited for smaller input sizes, such as MobileNet, ShuffleNet, or EfficientNet.
2. Differentiating Effects in the Co-Culture System: Since a co-culture system was used, it is crucial to distinguish the effects on the two different cell lines. The drug response could vary significantly between fibroblasts and epithelial cells, leading to distinct morphological features in each. The authors should make efforts to separate and analyze these effects individually.
3. Quality of Single-Cell Segmentation: Given the low resolution (10x, with 2x2 binning) of the images, the single-cell segmentation might not be entirely optimal. The authors should address the quality of segmentation, as this could be a critical factor affecting the accuracy of MOA predictions.
4. Exploration of Segmentation-Independent Approaches: Considering the resolution limitations, the authors should explore alternative approaches, such as segmentation-independent tiling. While this method may not be applicable for CellProfiler-based feature extraction, it could provide better inputs for NN-based approaches and potentially improve the overall analysis.

Minor comments:

1. The CK and vimentin images from the OVCAR8-WI38 co-culture in Supplementary Figure 1 appear very similar or identical. Could the authors clarify why this is the case?
2. While using MOA for performance comparison is a standard approach, the authors should consider exploring alternative methods, such as chemical similarity or the Connectivity Map, to provide additional insights.

3. The study highlights significant variability in the NN's performance across different co-culture conditions, which raises concerns about the generalizability and robustness of the NN-based approach, particularly when applied to heterogeneous datasets

Version 1:

Reviewer comments:

Reviewer #2

(Remarks to the Author)

Authors have addressed all review points, conducted additional experiments and substituted the main model. I don't have more remarks or questions and recommend the manuscript for publication.

Reviewer #3

(Remarks to the Author)

The initial manuscript predominantly focused on utilizing the ResNet50 neural network. However, the revised version broadens this scope by incorporating EfficientNetB0 and MobileNetV2, enhancing the diversity of neural network architectures used for feature extraction. Furthermore, the updated manuscript provides in-depth comparisons of enrichment scores across these networks, detailing specific percentages and addressing variations caused by different drug concentrations and the cellular/genetic contexts of the treatments. It also expands the discussion on methodological approaches, especially the challenges and solutions related to image analysis and feature extraction in co-culture assays. Enhanced by new figures and supplementary tables, the manuscript now offers a more detailed visual and quantitative analysis, including activation maps and PCA plots that better elucidate the findings. Additionally, a more rigorous evaluation of technical quality, such as Z' scores to assess assay quality, and refined feature extraction strategies better accommodate the complexities of co-culture data. These revisions indicate a comprehensive and profound advancement in the research methodologies and implications of feature extraction in cancer cell line co-cultures, demonstrating significant development in the depth and breadth of the revised manuscript.

All previously raised concerns have been satisfactorily addressed in the revised manuscript, and I have no further questions at this time.

Evaluating Feature Extraction in Ovarian Cancer Cell Line Co-Cultures Using Deep Neural Networks

Osheen Sharma^{1*}, Greta Gudoityte¹, Rezan Minozada¹, Olli P. Kallioniemi^{1,2}, Riku Turkki², Lassi Paavolainen^{2,3}, Brinton Seashore-Ludlow^{1**}

¹Department of Oncology-Pathology, Karolinska Institutet, Science for Life Laboratory, Solna, Sweden

²Institute for Molecular Medicine Finland (FIMM), HiLIFE, University of Helsinki, Finland

³iCAN Digital Precision Cancer Medicine Flagship, University of Helsinki, Finland

*Correspondence: osheen.sharma@ki.se

**Correspondence: brinton.seashore-ludlow@ki.se

Response to Reviewers

Reviewer #1

We thank the reviewer for taking the time to write the review. We agree and note that the journal we submitted to is the Communications Biology journal from Nature Portfolio.

Reviewer #2

1. For such small images (50x50 and 90x90), would it make more sense to use even smaller models, like ResNet 34\18?

Using smaller models can be advantageous, especially for datasets with smaller image sizes, as it helps to reduce computational costs and avoids overfitting. Based on the suggestions, we included feature extraction using two additional lightweight models, EfficientNetB0 and MobileNetV2, in the revised manuscript. These models were chosen because they have significantly fewer parameters (EfficientNetB0: 5.3M, MobileNetV2: 3.5M). The results are now changed to the ones computed by the EfficientNetB0 model and the remaining results from additional models are now included in the revised manuscript under the Results Section, and the performance of these models is summarized in Figure 3a-b, Figure 4c, Figure 5b-c, Supplementary Figure 7, 11, Table 2, Supplementary Table 4-6.

2. Masked single-cells vs unmasked seem to be complementary in some cases. For example, PARP AND BET seem to work better for O8W and KW (supp. figures 8,11). Would it make sense to concat \ pool features from surroundings and the cell?

We used CellProfiler to extract features from single-cell objects, where each cell is handled individually, focusing only on the morphological properties of the target cell. In contrast, bounding box-based neural network feature extraction from unmasked images inherently includes contextual information from surrounding cells in addition to the target cell, making the features incomparable to CellProfiler outputs. To ensure comparability, we masked the bounding box to isolate the single-cell object, aligning the neural network's focus with CellProfiler's single-cell feature extraction approach. While pooling or concatenating features from both the cell and its surroundings would reintroduce contextual information, making it similar to the unmasked approach, a more appropriate method would involve introducing weights or other strategies to combine these feature

vectors effectively¹. We consider this approach outside the scope of the current study but recognize it as an intriguing direction for future work.

3. Could you clarify the following: when calculating the enrichment score, was the query treatment in other concentrations excluded as a potential response?

In the original manuscript, the enrichment score calculations did exclude the query compound well but not other concentrations of the query compound. In the revised manuscript, we have updated the enrichment score calculation with both the other concentrations of the query compound as well as excluding the other concentrations of the query compound. This effectively lets us compare whether it is the query compound itself that is driving the enrichment. The results of this updated analysis are now included in the revised manuscript, in Table 2 (which includes all concentrations), Supplementary Table 4 (which includes all concentrations), and Supplementary Table 5 (excluding concentration from query compound).

4. Do you think it would make sense to report ES per drug per concentration to see at which concentration different drugs make a visible impact on the cell? Can it be the case that CellProfiler and NN detect drug effects at different concentrations?

We thank the reviewer for this constructive suggestion. We have now included supplementary data reporting enrichment score calculations for each co-culture combination, broken down by drug and concentration (e.g. per well correlations).

Additionally, we analyzed the enrichment scores across different concentration ranges by computing significance (p-values < 0.05) for the lowest two concentrations and the highest three concentrations. These results are now in Supplementary Table 6 and discussed in the revised manuscript, lines 382-394.

5. Lines 230-238: Can it be the case that PI3K-targeting drugs just kill the cells, similarly to other drugs which results in phenotypic inconsistency? Drugs of which target pop up at the top of the list if the PI3K-targeting drug was used as a query for enrichment analysis?

We understand the concern of the reviewer, and this is why we compare the DSS scores from monocultures using CellTiterGlo and nuclei count from the co-culture experiments. When we compare across cell lines, we see that PI3K MOA recovery varies across cell lines despite similar DSS. This suggests a similar response in terms of viability as demonstrated by the DSS scores but differences between the cell lines in demonstrating consistent phenotypic responses. We have rewritten this section to clarify the message (line 595-613).

6. Is there a correlation between the distance of consensus treatment profiles from consensus negative control profiles and cancer cell count and DSS?

DSS is calculated using cell count for the co-cultures. Higher DSS suggests higher cell toxicity across a concentration response of the compound. Thus, these are correlated. The negative controls are used in the DSS calculation. For more information, please see reference 34, Yadav et al. We have tried to clarify in the text several cases where DSS either underscores the observed MOA enrichments or deviates from the observed enrichment. We believe that these cases are useful in understanding the potentially confounding factors in each feature extraction approach, and when unexpected results are obtained.

7. Could you clarify whether the initial (not enhanced) dataset used all concentrations or only one per treatment?

The initial enrichment score calculation used all concentrations per treatment. However, in the revised manuscript, we have also included enrichment score calculations after removing the query compound and its other concentrations to ensure a robust and faithful comparison between feature extraction methods. These updated results are now detailed in the revised manuscript and Supplementary Table 5.

8. Do you think that your work would benefit if you tried some explanation technique (GradCAM)? For instance, with that, you could potentially observe the RoIs for the NN if the neighbouring cells are in the crop and what changes if not.

We explored this by plotting activation maps (Supplementary Figure 8) to better understand the feature extraction process. For unmasked bounding boxes, the activation maps revealed high-intensity activations in neighboring cells, indicating that the network likely incorporates contextual information from surrounding regions. While this can be useful in some cases, it does not align with the purpose of our study, which is focused on extracting features specific to individual cells. On the other hand, the masked bounding boxes showed activations concentrated on the single-cell region of interest, making them more suitable for our study and providing better comparability with CellProfiler.

9. I see that you have used trained models. I think you could include a baseline of ImageNet-pretrained network without further training.

We have tried to significantly clarify which models we are using throughout the manuscript. Initially, we examined baseline ImageNet pretrained models using a transfer learning approach (ResNet50, EfficientNetB0, MobileNetV2). The results from these experiments are presented in Table 2, Supplementary Table 4-6. Further, we fine-tuned EfficientNetB0 model for the MHB cell line to provide a comparative analysis. The results from these experiments are presented in Supplementary Table 7. Details of the feature extraction process for these models are described in the manuscript under methods section line 1143-1204.

Reviewer #3

Major Comments

- 1. Optimization of the Neural Network (NN) for Feature Extraction: Although using ResNet50 for feature extraction is a standard approach, it may require further optimization in this context. ResNet50 is pre-trained on larger tiles (224x224 pixels), which might not be ideal for smaller images (50x50 or 90x90 pixels). The authors should consider exploring other models better suited for smaller input sizes, such as MobileNet, ShuffleNet, or EfficientNet.**

We thank the reviewer for this suggestion. In the revised version of the manuscript, we have explored additional lightweight models better suited for smaller input image sizes, including MobileNetV2 and EfficientNetB0. These models were selected due to their significantly smaller parameter sizes and lesser convolutional layers (MobileNetV2: 3.5M, EfficientNetB0: 5.3M) compared to ResNet50 (25.6M), making them more suitable for the 50x50 and 90x90 image sizes used in this study. The results of feature extraction using these models have been included in the revised manuscript. Details of this analysis can be found in the Results Section, and the performance of these models is summarized in Figure 3a-b, Figure 4c, Figure 5b-c, Supplementary

Figure 7, 11, Table 2, Supplementary Table 4. In addition, we have fine-tuned the EfficientNetB0 model specifically for the MH & BjhTERT (MHB) co-culture combination because CellProfiler showed better overall recovery of MOA for this particular cell line (results shown in Supplementary Table 7). The results are discussed under the results section “*Enhanced training dataset through combined co-culture wells*”.

- 2. Differentiating Effects in the Co-Culture System: Since a co-culture system was used, it is crucial to distinguish the effects on the two different cell lines. The drug response could vary significantly between fibroblasts and epithelial cells, leading to distinct morphological features in each. The authors should make efforts to separate and analyze these effects individually.**

In this paper, we focused on analyzing the drug-induced morphological changes in ovarian cancer cells, as these are the primary targets of the treatments under investigation. While fibroblasts are also treated and could therefore exhibit drug-induced morphological changes, we aimed to evaluate how the presence of fibroblasts influences cancer cell morphology, not the fibroblasts themselves. Adding fibroblasts in culture provides insight into cancer cell behavior in a more physiologically relevant co-culture environment, which includes the impact of fibroblast-cancer cell interactions. By isolating cancer cell analysis, we maintain focus on the primary objective of understanding drug responses in the cancer cells.

- 3. Quality of Single-Cell Segmentation: Given the low resolution (10x, with 2x2 binning) of the images, the single-cell segmentation might not be entirely optimal. The authors should address the quality of segmentation, as this could be a critical factor affecting the accuracy of MOA predictions.**

We agree that accurate segmentation is a critical step for reliable analysis and MOA predictions. In this study, we used Cellpose, a robust and widely used deep learning model specifically trained for cellular segmentation tasks. To ensure the quality of segmentation, the results were carefully reviewed and validated by additional experts in our lab.

- 4. Exploration of Segmentation-Independent Approaches: Considering the resolution limitations, the authors should explore alternative approaches, such as segmentation-independent tiling. While this method may not be applicable for CellProfiler-based feature extraction, it could provide better inputs for NN-based approaches and potentially improve the overall analysis.**

While tiling could be an alternative to segmentation-based approaches, it is not suitable for our study due to the co-culture system. Our analysis focuses on the drug-induced effects on cancer cells, requiring a method to distinguish between the two cell types. Without segmentation, it is not possible to handle the two cell types individually, which is essential for isolating the morphological changes specific to the cancer cells. Segmentation remains a critical step for achieving our study's objectives and ensuring accurate cell-type-specific analysis.

Minor Comments

- 1. The CK and vimentin images from the OVCAR8-WI38 co-culture in Supplementary Figure 1 appear very similar or identical. Could the authors clarify why this is the case?**

Our staining approach is based on epithelial and mesenchymal cell markers. Epithelial cells, such as ovarian cancer cells, are marked with CK8/18, and mesenchymal cells, like fibroblasts, are marked with Vimentin. However, ovarian cancer cells may exhibit epithelial-mesenchymal transition (EMT), especially under aggressive phenotypes or metastatic potential. As a result, OVCAR8 cells can express both CK8/18 and Vimentin. Additionally, interactions between cancer cells and fibroblasts may influence gene expression, potentially increasing Vimentin expression in cancer cells. These factors explain the overlap observed in the staining patterns. We also observed the same pattern when we did screening on OVCAR8 cocultured with BjhTERT (this assay is not included in the study, shown in Supplementary Figure 15).

2. While using MOA for performance comparison is a standard approach, the authors should consider exploring alternative methods, such as chemical similarity or the Connectivity Map, to provide additional insights.

We thank the reviewer for this constructive comment and agree that chemical similarity within particular MoAs may be confounding. This study aimed to compare feature extraction methods and evaluate their ability to capture morphological features across ovarian cancer cell lines treated with drugs at varying concentrations. While chemical similarity and Connectivity Map analyses offer valuable insights, they fall outside the scope of this work. We are addressing the chemical similarity approach in a follow-up project to further explore compound space and phenotypic relationships.

3. The study highlights significant variability in the NN's performance across different co-culture conditions, which raises concerns about the generalizability and robustness of the NN-based approach, particularly when applied to heterogeneous datasets

We hypothesize that CellProfiler effectively captures features even from lower-resolution images, while neural networks are more sensitive to image resolution. Since this dataset was acquired at 10x magnification with 4 fields of view, NN performance may be impacted by resolution. However, EfficientNetB0 showed consistent performance across co-culture combinations, and training ResNet50 on a pooled dataset (middle 3 concentrations) further improved performance. This indicates that while pretrained models initially varied, they were still comparable to CellProfiler in extracting relevant features. Additionally, the 5-point drug concentration design may influence results, as lower concentrations are less likely to induce strong morphological changes.

References

1. Toth, T., Bauer, D., Sukosd, F. & Horvath, P. Fisheye transformation enhances deep-learning-based single-cell phenotyping by including cellular microenvironment. *Cell Reports Methods* 2, (2022).